# Rapid Authentication of Intact Stingless Bee Honey (SBH) by Portable LED-Based Fluorescence Spectroscopy and Chemometrics

**DOI:** 10.3390/foods13223648

**Published:** 2024-11-16

**Authors:** Diding Suhandy, Dimas Firmanda Al Riza, Meinilwita Yulia, Kusumiyati Kusumiyati, Mareli Telaumbanua, Hirotaka Naito

**Affiliations:** 1Department of Agricultural Engineering, Faculty of Agriculture, The University of Lampung, Jl. Soemantri Brojonegoro No.1, Bandar Lampung 35145, Indonesia; mareli.telaumbanua@fp.unila.ac.id; 2Department of Biosystems Engineering, Faculty of Agricultural Technology, University of Brawijaya, Jl. Veteran, Malang 65145, Indonesia; dimasfirmanda@ub.ac.id; 3Department of Agricultural Technology, Lampung State Polytechnic, Jl. Soekarno Hatta No. 10, Rajabasa, Bandar Lampung 35141, Indonesia; meinilwitayulia@polinela.ac.id; 4Department of Agronomy, Faculty of Agriculture, Universitas Padjadjaran, Sumedang 45363, Indonesia; kusumiyati@unpad.ac.id; 5Graduate School of Bioresources, Department of Environmental Science and Technology, Mie University, 1577 Kurima-machiya-cho, Tsu 514-8507, Mie, Japan; naito@bio.mie-u.ac.jp

**Keywords:** adulteration, brown rice syrup, chemometrics, honey authentication, portable fluorescence spectroscopy, stingless bee honey

## Abstract

Indonesian stingless bee honey (SBH) of *Geniotrigona thoracica* is popular and traded at an expensive price. Brown rice syrup (RS) is frequently used as a cheap adulterant for an economically motivated adulteration (EMA) in SBH. In this study, authentic Indonesian *Geniotrigona thoracica* SBH of *Acacia mangium* (*n* = 100), adulterated SBH (*n* = 120), fake SBH (*n* = 100), and RS (*n* = 200) were prepared. In short, 2 mL of each sample was dropped directly into an innovative sample holder without any sample preparation including no dilution. Fluorescence intensity was acquired using a fluorescence spectrometer. This portable instrument is equipped with a 365 nm LED lamp as the fixed excitation source. Principal component analysis (PCA) was calculated for the smoothed spectral data. The results showed that the authentic SBH and non-SBH (adulterated SBH, fake SBH, and RS) samples could be well separated using the smoothed spectral data. The cumulative percentage variance of the first two PCs, 98.4749% and 98.4425%, was obtained for calibration and validation, respectively. The highest prediction accuracy was 99.5% and was obtained using principal component analysis–linear discriminant analysis (PCA-LDA). The best partial least square (PLS) calibration was obtained using the combined interval with R^2^_cal_ = 0.898 and R^2^_val_ = 0.874 for calibration and validation, respectively. In the prediction, the developed model could predict the adulteration level in the adulterated honey samples with an acceptable ratio of prediction to deviation (RPD) = 2.282, and range error ratio (RER) = 6.612.

## 1. Introduction

In Indonesia, stingless bee honey (SBH) is mostly collected from several bee species such as *Heterotrigona itama*, *Tetrigona apicalis*, and *Geniotrigona thoracica*. Among them, *Geniotrigona thoracica* is the largest and the most popular highly prized SBH and is traded at an expensive price. It has economic significance for most beekeepers in Indonesia. Due to its exceptional nutritional benefits and excellent biological properties, SBH is in high demand, and it was so especially during the COVID-19 pandemic [1,2]. According to Yaacob et al. [3] and Biluca et al. [4], SBH’s phenolic profiles are primarily responsible for its biological properties. Catechin, epicatechin, protocatechuic acid, p-coumaric acid, chlorogenic acid, and rutin are among the significant flavonoids and phenolic acids that have been identified in SBH as having biologically active effects [5,6]. According to earlier research, the antioxidant activity and total phenolic and flavonoid content of SBH are considerably higher than those of *Apis mellifera* and *Apis dorsata* [6,7]. However, the limitation of its production has raised concerns about SBH adulteration in the commercial market [8]. It is reported that, currently, honey ranks third among the most adulterated foods in the world [9]. Adulterating SBH by intentionally adding artificial industrial syrups (corn, sugarcane, beet, wheat, tapioca, and rice syrups) and mislabeling SBH with fake honey are food frauds that frequently occur in commercial honey markets. Syrup adulteration not only enhances the flavor of honey but also lowers its nutritional value and, in severe conditions, may lead to certain chronic diseases. The adulterated syrup in SBH raises blood sugar levels, which can lead to excessive blood pressure, blood lipid levels, diabetes, and abdominal weight gain [10]. Adulterants may also harm internal organs, increasing visceral fat pads and total body fat, which can result in mortality [11,12], causing a fatty liver [13], and acute and chronic kidney injury [14]. Brown rice syrup (RS) with similar color properties to most SBH is available in abundance and is frequently used as a cheap adulterant for economically motivated adulteration (EMA) in SBH [15,16,17,18].

Traditionally, SBH authentication involves expensive and time-consuming analytical methods based on a stable carbon isotope ratio analysis (SCIRA) (^13^C/^12^C = δ^13^C (‰)) [17,19,20,21]. This method is effective in detecting the existence of artificial syrup from C4 plants (plants with the Hatch–Slack photosynthesis cycle) and CAM plants (plants with the Crassulacean Acid Metabolism cycle) added intentionally into SBH due to different ranges of δ^13^C values even at low levels of C4 syrup adulteration (more than 7%) [22]. Nevertheless, it requires a highly trained operator to perform the analysis [23,24]. SCIRA also has limitations. According to Wang et al. [25], this approach is not appropriate for high-throughput routine screening for honey adulteration because it typically necessitates lengthy analysis times and costly equipment. Additionally, RS and most SBHs belong to the C3 plants cycle (the Calvin and Benson cycle), thus making adulteration of SBH with RS more challenging because the difference in δ^13^C‰ between RS and SBH is too small to be detected [26]. As a result, false positives have been reported with honey produced by bees that may have fed on C4 plants analyzed by SCIRA [27]. For this reason, applying official analysis methods based on SCIRA for honey authentication is impossible for detecting RS adulteration in most SBH [28]. 

Several non-targeted advanced analytical methods based on UV–visible (UV–vis) spectroscopy to far infrared spectroscopy have been applied to identify and quantify honey adulteration with various artificial syrups [8,9,15,16,18,19,20,21,23,24,28,29,30,31]. At a low adulteration level below 10% (*w*/*w*), infrared spectroscopy showed a good detection performance, and a ratio of prediction to deviation (RPD) of more than 2.2 was obtained [32]. Full-spectrum PLS regression (FPLSR) coupled with spectroscopy is widely used in chemometrics to determine the honey adulteration level as reported by current works [8,32]. However, the typical analytical information from FPLSR overlapped and had more interference problems, resulting in a lower prediction performance. For this reason, the performance of FPLSR was effectively improved with several different variable selections to quantify the honey adulteration level. Peng et al. [33] used laser-induced breakdown spectroscopy and PLS with three feature selection methods, including genetic algorithm (GA), variable importance in projection (VIP), and the selectivity ratio (SR), to quantify the corn syrup adulteration level in Acacia honey. Li et al. [34] combined mid-infrared (MIR) and Raman data with interval PLS (iPLS) for quantitative analysis of honey adulteration. NIR spectroscopy both using benchtops and portable or handheld devices is rapid, non-destructive, and accurate. However, instrumentation is still less affordable to most developing countries, with Indonesia not being an exception. On the other hand, a benchtop UV–visible spectroscopy in the range of 190–700 nm with more affordable instrumentation is a promising analytical method for honey authentication, as reported by many previous works with acceptable results [8,35,36,37,38,39,40,41,42,43]. However, UV–vis spectroscopy is less sensitive and laborious with dilution procedures still required for highly concentrated honey samples [44]. 

Fluorescence spectroscopy with 100-to-1000-fold higher sensitivity than UV–vis spectroscopy has been explored for honey authentication purposes [21,44,45,46]. Several important chemical compounds of SBH, such as aromatic amino acids, water-soluble vitamins (riboflavin, nicotinic acid, pantothenic acid, folic acid, and ascorbic acid), and many polyphenols (such as chlorogenic acid, p-coumaric acid, gallic acid, vanillic acid, and caffeic acid), have fluorescence information [47]. However, traditional benchtop fluorescence spectroscopy equipped with a quartz cuvette as a sample holder leads to difficulty in properly acquiring fluorescence spectral data for high-absorbance solutions such as honey [48]. Dilution is a common and simple way to solve this problem. Additionally, it involves the measurement of the EEM (excitation–emission matrix) using a benchtop fluorescence spectrometer, which is expensive, time-consuming, and unsuitable for field measurement. Handling EEM data is also not easy and requires a complex chemometric calculation. For this reason, Suhandy et al. [49] proposed, for the first time, the use of portable fluorescence spectroscopy equipped with a light-emitting-diodes (LED) lamp as a fixed excitation source as a cost-effective, and acceptable method to detect and quantify high-fructose corn syrup (HFCS) adulteration in SBH. The result is promising with a coefficient of determination in prediction (R^2^_pred_) = 0.9627 and root mean squared error of prediction (RMSEP) = 4.1579%. However, that previous study involved using a dilution treatment and a quartz cuvette for the sample holder, making measurement still slow and laborious. It is reported that the dilution ratio of honey and quartz cuvette highly affected the quality of fluorescence intensity [50]. To solve this problem, in the present study, we proposed a novel method based on portable LED-based fluorescence spectroscopy equipped with an innovative sample holder to rapidly detect and quantify rice syrup (RS) adulteration in Indonesian SBH. 

## 2. Materials and Methods

### 2.1. Samples

In the present study, 100 Indonesian monofloral pure authentic SBH samples of *Geniotrigona thoracica* were harvested and processed in early 2024 from PT Suhita Lebah Indonesia, a local trusted SBH producer located in Lampung Province, Sumatra, Indonesia (5°27′ S and 105°16′ E latitude and longitude, respectively, with an altitude of 100 m). The nectar source of the SBH sample is mainly *Acacia mangium* with a light-yellow color, as seen in Figure 1. The *Acacia mangium* tree is one of the most popular and fast-growing tree species in Indonesia and in many other parts of Asia. To perform a reliable authentication test, we provide three non-SBH samples: 100 samples of pure fake SBH, 120 samples of adulterated SBH, and 200 samples of pure brown rice syrup (RS). Fake honey samples were commonly produced by feeding bee colonies with a synthetic honey solution, which bees later converted into artificial honey. An undisclosed supplier provided fake honey samples for this research. RS was purchased from the local traditional market at Bandar Lampung City, Lampung Province, Indonesia. Adulterated SBH was intentionally created by adding brown rice syrup (RS) to pure authentic SBH samples at low (10% and 20%), medium (30% and 40%), and high (50% and 60%) adulteration levels (*v*/*v*). This adulteration level (10–60%) is in the range of EMA adulteration and is similar to previous works [36,37]. It is noted that the color of the adulterated SBH becomes darker as the level of adulteration increases, as depicted in Figure 2. It has been noted that the color of honey varies according to the amounts of anthocyanins, beta-carotene, and chlorophyll and its derivatives. Consequently, honey’s color is greatly influenced by its botanical origin [51,52]. Therefore, it is unreliable to distinguish between pure and adulterated SBH samples based only on color information, particularly at low adulteration levels (10–20%). According to Julika et al. [53], SBH stored at room temperature (25 °C) deteriorates faster than at 4 °C. For this reason, all samples were stored in a domestic refrigerator (at 4 °C) for 1 day before spectral acquisition.

### 2.2. Fluorescence Spectral Data Acquisition

All samples were heated in a water bath at 60 °C for 30 min before spectral acquisition [49,54]. For each sample, 2 mL of authentic SBH, fake SBH, adulterated SBH, and RS was dropped directly into an innovative sample holder without dilution. Fluorescence spectral data were measured using a portable fluorescence spectrometer in front-face mode equipped with a 365 nm LED lamp as the fixed excitation light source in the range of 348.5–866.5 nm with a 0.5 nm increment, as seen in Figure 3. The front-face mode or reflectance mode is appropriate and widely used for honey fluorescence measurement [55,56,57,58]. A computer equipped with SpectroLab^®^ software (version 1.0.24) was connected to the spectrometer (GoyaLab, IndiGo Fluo UV spectrometer, Talence, France) via a USB cable to control the spectral acquisition, and the following parameters were used: 2000 ms of exposure time, 100 ms of delay time between cycles, and 10 cycles. Spectral acquisition was performed at room temperature (26–27 °C).

### 2.3. Data Analysis

The relationship between fluorescence spectral data as the independent variables (predictors) and type of honey as the dependent variable (target) was qualitatively evaluated by using principal component analysis (PCA). PCA was calculated based on the non-linear iterative partial least squares (NIPALS) algorithm. Information from the scores and x-loading plot was utilized to check for possible discrimination between four different samples (authentic SBH, adulterated SBH, fake SBH, and RS) based on fluorescence spectral data. Information from an influence plot (Hotelling’s T^2^ and Q-residual) was used to evaluate the possible occurrence of outliers. To enhance spectral data difference and improve the signal-to-noise ratio (SNR), data preprocessing of moving averaging smoothing (MAS) with 21 segments of smoothing points was applied. PCA was calculated for the raw and smoothed spectral data. Four popular and widely used supervised classification methods based on soft independent modeling of class analogy (SIMCA), partial least square–discriminant analysis (PLS-DA), linear discriminant analysis (LDA), and principal component analysis–linear discriminant analysis (PCA-LDA) were calculated using smoothed spectral data for performance comparison. Recently, those classification methods were extensively utilized for honey adulteration [59,60]. The acceptability of the classification model was evaluated according to the accuracy parameter calculated as the ratio between the number of correct classified samples and the total number of samples [49,61,62]. For more conceptual information on the classification methods used in the present study, readers can be referred to several previous works [63,64,65,66].

To perform quantitative analysis, a calibration model for estimating the adulteration level was developed by using partial least square regression (PLSR) with full-spectrum and selected intervals. PLSR is well known as a standard linear calibration model for quantitative analysis using full-spectrum spectral data [67]. Interval PLS (iPLS) with fewer input variables enabled the improvement in the full-spectrum PLS for the quantification of food adulteration, provided better performance, faster measurement, and reduced the cost [68]. Currently, iPLS is reported for honey authentication and that of other foods [69,70,71]. Two important parameters of the ratio of prediction to deviation (RPD) and range error ratio (RER) were used for calibration model evaluation. The minimum acceptable value for rough screening application is higher than 2.0 and higher than 4.0 for RPD and RER, respectively [72]. The Unscrambler X (version 10.5, CAMO, Norway) was used for all chemometrics calculations. 

## 3. Results and Discussion

### 3.1. Spectral Analysis of SBH and Non-SBH Samples

The typical raw and smoothed fluorescence spectral data of authentic SBH and non-SBH (adulterated SBH, fake SBH, and RS samples) are depicted in Figure 4. The data presented were in line with previous reported studies [21,45,49,73,74]. Four samples that have identical spectral shapes with three distinct peaks at 365 nm, 476 nm, and 721 nm could be observed. In general, the fluorescence spectral data obtained in the present study could be related to some important compounds in honey such as niacin (vitamin B_3_), caffeic acid, and sinapic acid [75]. RS samples have the highest fluorescence intensity, while the fake SBH has the lowest one. Previously, Hao et al. [21] reported the fluorescence measurement of corn syrup adulteration with a fixed excitation wavelength at 230 nm and 280 nm. One major fluorescence peak was observed at 340 nm and it is related to the presence of aromatic amino acids and non-flavonoid phenolic compounds in honey. Strong emission at 476 nm could be closely related to the fluorescence peak of phenolic substances (folic acid or vitamin B_9_), as reported by several previous studies [45,49,58,76,77]. Banaś and Banaś [45] utilized three fixed excitation wavelengths at 250 nm, 290 nm, and 375 nm (λ_ex_ 250 nm, λ_ex_ 290 nm, and λ_ex_ 375 nm) to characterize several herb honeys. A major peak at 450 nm was identified at 375 nm of excitation. A minor peak at 365 nm was noisy and it is related to a small fraction of the excitation wavelength used in this study (λ_ex_ 365 nm). Another minor peak at 721 nm could be related to the frequency-doubled peak (FDP) phenomenon as reported by Yan et al. [74]. It is seen that the FDP of the four samples is significantly different. The FDP intensity at 721 nm is highly affected by the type of sample and may serve as a potential marker to discriminate authentic SBH from non-SBH samples (adulterated SBH, fake SBH, and RS).

To evaluate the influence of adding rice syrup on the quality of authentic SBH spectral data, a plot of smoothed fluorescence spectral data of adulterated SBH with three adulteration levels (low, medium, and high) is presented in Figure 5. It is observed that when different proportions of rice syrup (10–60%) were adulterated into authentic SBH, the fluorescence intensity decreased with an increase in the rice syrup concentration, and the peak positions of the adulterated honey samples at a wavelength of 375 nm were also redshifted. This phenomenon is also observed in previous similar studies [21].

### 3.2. Principal Component Analysis

The PCA score plot of the four samples (authentic SBH, adulterated SBH, fake SBH, and RS) is demonstrated in Figure 6. The cumulative percentage variance of the first two principal components, 98.41% and 98.48%, was obtained for raw and smoothed spectral data, respectively. Based on PCA calculation on raw spectral data, the discrimination between authentic SBH and non-SBH samples (adulterated SBH, fake SBH, dan RS) could be established using the PC1-axis, as seen in Figure 6a. The result of the PCA score plot shows that a separation of SBH and non-SBH samples was achieved. SBH samples and fake SBH were on the left of PC1 (negative PC1 scores). The RS samples were clustered on the right of PC1 with positive scores. The adulterated SBH (mixture of authentic SBH and RS) was clustered between authentic SBH and RS samples in the positive and negative PC1 scores. Better separation of authentic SBH and non-SBH samples was achieved using smoothed spectral data, as seen in Figure 6b. The adulterated SBH samples with low adulteration levels (10%, and 20%) were closely clustered to the authentic SBH on the left of PC1. The medium and high levels of SBH sample adulteration (30%, 40%, 50%, and 60%) were on the right of PC1 and close to the RS cluster.

The x-loading plot shows that the wavelengths at 476 nm and 721 nm have major positive loading at PC1 and PC2, respectively, and play an important role in the separation between SBH and non-SBH samples, as can be seen in Figure 7. A minor peak at 721 nm with positive loading is also observed at PC1. The x-loading plot results confirmed that the separation of SBH and non-SBH samples in the present study is mainly due to the phenolic substance difference and FDP phenomenon. To develop robust supervised classification methods, a Hotelling’s T^2^ and Q-residual statistics plot of all authentic SBH and non-SBH (adulterated SBH, fake SBH, and rice syrup) samples (*n* = 520) was calculated at 95% confidence and is presented in Figure 8. Hotelling’s T^2^ is a standard approach to determine the significance of multivariate distances, while Q-residual represents how well samples are assessed by the model [40]. It is an influence plot used to identify and remove the anomalous fluorescence spectral data that are considered outliers [78]. According to Figure 8, there are no outliers detected as no samples exceeded the thresholds (blue and red dashed lines for Hotelling’s T^2^ and Q-residual, respectively). 

### 3.3. Results of Classification: Model Development

To develop supervised classification methods, all samples (*n* = 520) were randomly separated into two different sets: 60% belong to calibration and validation sample set (*n* = 60, *n* = 72, *n* = 60, and *n* = 120 for authentic SBH, adulterated SBH, fake SBH, and rice syrup (RS) classes, respectively), and 40% belong to prediction sample set (*n* = 40, *n* = 38, *n* = 40, and *n* = 80 for authentic SBH, adulterated SBH, fake SBH, and rice syrup (RS) classes, respectively). The calibration and validation sample set was used to develop a SIMCA model for each class, as shown in Table 1. The optimum number of principal components (PCs) used for each class was determined by using a full-cross-validation method. As seen in Table 1, a SIMCA model for each class was constructed with a different number of optimum PCs. Three PCs were used to construct an authentic SBH model with the obtained CPV in calibrations and validation of 98.3490%, and 98.1350%, respectively. Four PCs were used to develop SIMCA models for adulterated SBH, fake SBH, and rice syrup class with the obtained CPV in the calibration of 99.3491, 99.0731, and 98.8600%, respectively. 

To provide better classification, several studies have shown the extensive application of the PLS-DA, LDA, and PCA-LDA methods for qualitative studies [79]. In this study, PLS-DA was employed for the full spectrum in the range of 348.5–866.5 nm and it was calculated based on the NIPALS algorithm and a full cross-validation at a 95% confidence level. The result was acceptable with accuracy =79.17%, and 78.21% for calibration and validation, respectively. The coefficient of determination (R^2^) was also high with R^2^ = 0.89, and R^2^ = 0.88 for calibration and validation, respectively. Several works reported comparable accuracy and R^2^ of the PLS-DA model for honey authentication. For example, Matwijczuk et al. [80] developed a PLS-DA model for the classification of honey powder with R^2^ = 0.89~0.91. David and Magdas [24] used Raman spectroscopy and the PLS-DA method to authenticate the honey origin and harvesting year. The accuracy was 81.13~97.16% depending on the spectral range and preprocessing methods. 

The result of LDA and PCA-LDA model development is presented in Figure 9. LDA and PCA-LDA are supervised classification methods where the number of variables is smaller than the number of samples. The variable selection for LDA was performed based on an x-loading plot. For this, we selected two wavelengths with high x-loadings as input variables: 476 and 721 nm. In PCA-LDA, 10 principal components were used as input variables in the range of 348.5–866.5 nm. According to Figure 9, the accuracy obtained for LDA is 86.22%. A better result was obtained for PCA-LDA. Most samples were separated into SBH and non-SBH categories. Figure 9 illustrates that some of the RS, adulterated SBH, and authentic SBH samples still overlap and cannot be distinguished using the established PCA-LDA model. This model had an accuracy of 98.08%, misclassifying five adulterated SBH samples as authentic SBH and one RS sample as an adulterated SBH sample. Compared to PLS-DA, it is noted that LDA and PCA-LDA with fewer input variables successfully improved the accuracy of the classification model. Previous studies also showed a superiority of LDA and PCA-LDA. Raypah et al. [81] studied the identification and quantification of SBH adulteration and showed that the developed PCA-LDA model could classify SBH and its adulteration with 100% accuracy. The PCA-LDA classification model is one of the most widely used for honey authentication. Nayik et al. [82] applied PCA-LDA to classify three Indian honeys based on antioxidant properties and sugar parameters, and reported 100% accuracy for individual and combined sample sets.

### 3.4. Result of Classification: Model Evaluation

The performance of the classification models was evaluated and the result is presented in Table 2. We used 208 samples (*n* = 40, *n* = 48, *n* = 40, and *n* = 80 for authentic SBH, adulterated SBH, fake SBH, and rice syrup, respectively). The SIMCA model has the lowest performance with 78.1% accuracy. The performance of the PLS-DA and LDA classification models was almost similar, with 86.5% and 85.6% accuracy, respectively. In the PCA-LDA model, only one sample of fake SBH was misclassified as adulterated SBH, resulting in the highest accuracy of 99.5% being obtained. Our result was comparable to previous results on honey adulteration detection using various spectroscopic and other methods. In a previous report, a rice syrup adulteration study using different spectroscopic methods, the obtained accuracy for honey authentication was 90~98% [83,84]. ATR-FTIR spectral data and four supervised classification models were utilized to detect adulteration of rice syrup in Chinese honey with different origins. The best accuracy (97.09%) was obtained for the first derivative-least squares support vector machines (Der-LS-SVM) [83]. Raman spectroscopy and three different classification methods (PLS-DA, support vector machine or SVM, and convolutional neural network or CNN) were applied to study honey adulteration with rice, corn, and maltose syrup. The CNN method was the best, with 99.76% accuracy being obtained [84]. Suhandy and Yulia [39] used UV spectroscopy with a benchtop spectrometer to classify Indonesian honey depending on botanical, entomological, and geographical origin. The result was acceptable, with 100% accuracy obtained using the SIMCA method. A similar spectroscopic method was used by Dimakopoulou-Papazoglou et al. [40] to detect adulteration in Mediterranean honeys with syrup and colorants. The accuracy was 98% using the PLS-DA classification model. NIR spectroscopy has been used for the detection of honey adulteration with corn and maltose syrups [85]. The PLS-DA model was developed using a selected wavelength and resulted in 86.3% and 96.1% accuracy for corn and maltose adulteration, respectively. Visible-NIR spectroscopy with the LDA method was combined to detect high-fructose corn syrup adulteration in Spain’s pure multi-floral honey and reported 100% accuracy [86]. Amiry et al. [87] utilized rheological and physicochemical parameters coupled with the LDA method to discriminate between pure and adulterated Iranian honey and reported 100% accuracy. Inverted sugar syrup in concentrations of 7%, 15%, and 30% was used as an adulteration level. Recently, a standard method for honey adulteration detection with sugar syrup (including corn and rice syrup) was developed using high-performance liquid chromatography with UV detection (HPLC-UV). The accuracy was 100% for the classification between pure and adulterated honey by PLS-DA [88]. 

### 3.5. Result of Quantification

In a quantitative study, we used 120 adulterated samples with 10–60% adulteration levels. The samples were divided into two sample sets: 60% of the samples (*n* = 72) were used for developing a calibration and validation model. The remaining samples (*n* = 48) were used for prediction purposes. Table 3 shows the model development results for quantifying brown rice syrup (RS) adulteration using PLS regression (PLSR) with a full spectrum and several selected intervals using the NIPALS algorithm and the full-cross-validation method. The full-spectrum PLS model involved 1037 variables (348.5–866.5 nm with an 0.5 nm increment). It is noted that the obtained RER values for the full-spectrum and interval PLS model meet the requirements for an acceptable PLS model. According to Parrini et al. [89], an RER between 3 and 10 and higher than 10 indicates moderate and good practical utility. Most developed calibration models have sufficient RPD values (RPD between 1.360 and 2.793). Williams and Sobering [90] suggested an accurate estimation capacity if the RPD values were higher than the limit of 2.5. Based on Table 3, the calibration model in intervals 3, 4, and 5 have RPD values higher than 2.5 and RER higher than 7.0. For this reason, these intervals were selected to develop the combined-interval PLS calibration model with 300 input variables. Figure 10 shows the scatter plot for the PLS calibration and validation results using full-spectrum and combined intervals. The full-spectrum and combined-interval PLS models used five latent variables and had similar performances, with R^2^_cal_= 0.899 and R^2^_val_= 0.874. However, the combined-interval PLS has a lower root mean square error of cross-validation (RMSECV = 6.138%). The RPD of the combined-interval PLS model is 2.802, higher than the RPD in the full-spectrum PLS model (RPD = 2.793). Our PLS model using a combined interval is better compared to that in previous work by Benković et al. [91]. They reported the best PLS model for quantification of glucose syrup adulteration in Acacia honey (10–90% adulteration level) using NIR spectroscopy with R^2^_cal_ of 0.8978, root mean square error of calibration (RMSEC) of 10.1552%, R^2^_cval_ of 0.8557, and RMSECV of 12.2011%. In a similar study using UV–vis and NIR spectroscopy, Valinger et al. [36] prepared pure and adulterated honey samples (10–90% adulteration level of high-fructose corn syrup) and reported a PLS model with the following characteristics: R^2^_cal_ = 0.9268, RMSEC = 8.5544%, R^2^_cval_ = 0.9100, RMSECV = 9.5683%, RPD = 3.3150, and RER = 10.4512. Our PLS model is superior compared to the PLS model coupled with the FTIR-ATR method. Anjos et al. [92] reported an acceptable result on the application of FTIR-ATR spectroscopy with PLS regression for honey adulteration with fructose and glucose. The best PLS calibration model could be generated with an R^2^ of 86.60 and 86.01 with an RPD of 2.6 and 2.55, respectively, for fructose and glucose.

The performance of the combined-interval PLS model in prediction was evaluated using 48 independent prediction samples and the result is presented in Figure 11. The result was acceptable with R^2^_pred_ = 0.804, root mean square error of prediction (RMSEP) = 7.562%, RPD = 2.282, and RER = 6.612. Through a 95% confidence pair *t*-test, there were no significant differences between the actual and predicted adulteration levels. Using a similar instrument equipped with a cuvette as a sample holder and diluted honey samples, Suhandy et al. [49] developed a calibration model for honey adulteration with HFCS (corn syrup). All models were satisfactory with an R^2^_pred_ greater than 0.90 for the PLSR, PCR, and MLR models. It is noted that the lower performance obtained in the present study was probably due to the similarity of authentic SBH and rice syrup as derived from the C3 (Calvin cycle) photosynthetic pathway.

Compared to other spectroscopy-based methods for honey adulteration with rice syrup, our prediction performance is similar and comparable. Ciursă et al. [93] utilized a benchtop Fourier-Transform Infrared (FTIR) spectroscopy and developed a PLS model for predicting various artificial syrups in Romanian honey with different botanical origins. The performance of PLS in prediction is R^2^ = 0.65~0.932 depending on the type of sugar. Chen et al. [94] used three-dimensional fluorescence spectroscopy with a costly benchtop spectrometer coupled with the PLSR method to quantify various levels of honey adulteration with rice syrup. The RPD in prediction was 3.05~12.37 depending on the type of honey. Visible and NIR spectroscopy were applied to study honey adulteration with glucose syrup. The adulteration level was 5–33% (*w*/*w*). The PLS was used for developing a calibration model for the quantification of the adulteration level and reported the RPD = 2.06 for prediction [95]. In the present study, portable LED-based fluorescence spectroscopy combined with simple PLS regression enabled rice syrup quantification with RPD = 2.282, demonstrating comparable performance to previous approaches while offering significant advantages associated with the speed and affordability of the technique.

## 4. Conclusions

This study has demonstrated that the adulteration of authentic SBH with non-SBH samples (adulterated SBH, fake SBH, and rice syrup) can be detected through the implementation of a rapid, simple, sensitive, and affordable method based on portable LED-based fluorescence spectroscopy combined with an innovative sampler holder and appropriate chemometric analysis. Both qualitative and quantitative results are acceptable. The PCA-LDA offered the best classification model with a calibration accuracy of 98.08% and a prediction accuracy of 99.5%. With a slight difference between calibration and prediction accuracy (1.42%), the PCA-LDA model with fewer variables typically produces a more robust classification model. With R^2^_pred_ = 0.804, RMSEP = 7.562%, RPD = 2.282, and RER = 6.612, the combined-interval PLS model with fewer variables demonstrated satisfactory prediction performance in predicting the adulteration level. Simple preprocessing of fluorescence spectral data combined with user-friendly chemometric methods enabled the implementation of this innovative technique for routine analysis in the honey industry to support food traceability and food safety.

## Figures and Tables

**Figure 1 foods-13-03648-f001:**
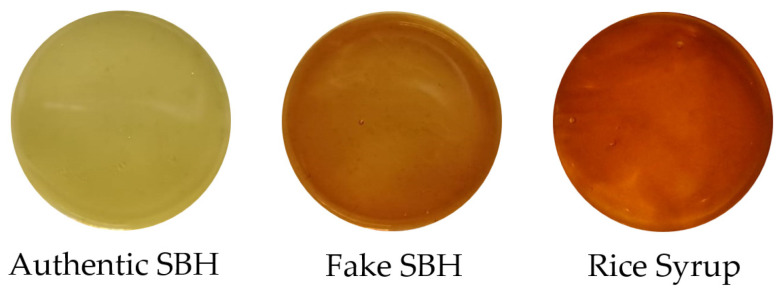
Visual information of authentic SBH, fake SBH, and brown rice syrup (RS).

**Figure 2 foods-13-03648-f002:**
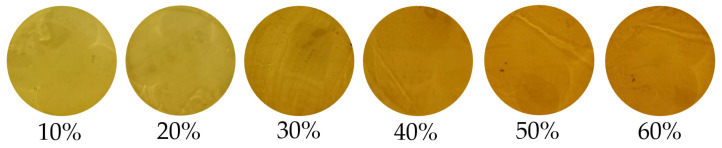
Visual information of adulterated SBH with six different adulteration levels.

**Figure 3 foods-13-03648-f003:**
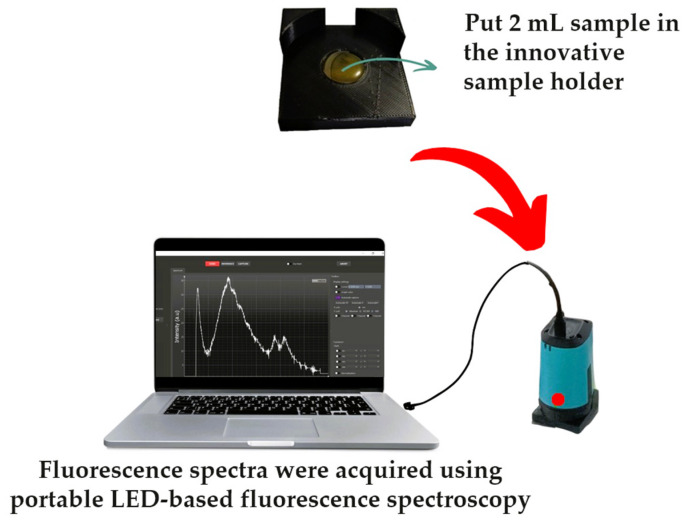
The front-face mode spectral acquisition system with portable LED-based fluorescence spectroscopy equipped with an innovative sample holder.

**Figure 4 foods-13-03648-f004:**
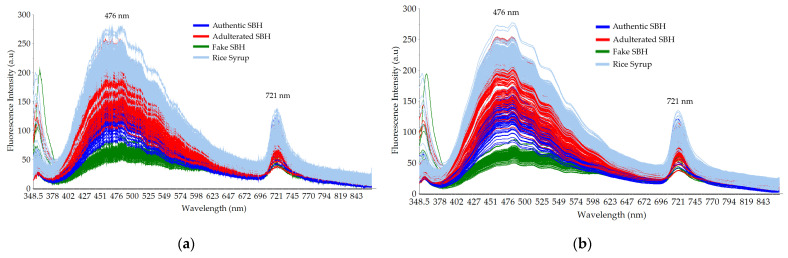
Typical fluorescence spectral data in the range of 348.5–866.5 nm with a fixed excitation at 365 nm: (**a**) raw spectral data; (**b**) smoothed spectral data.

**Figure 5 foods-13-03648-f005:**
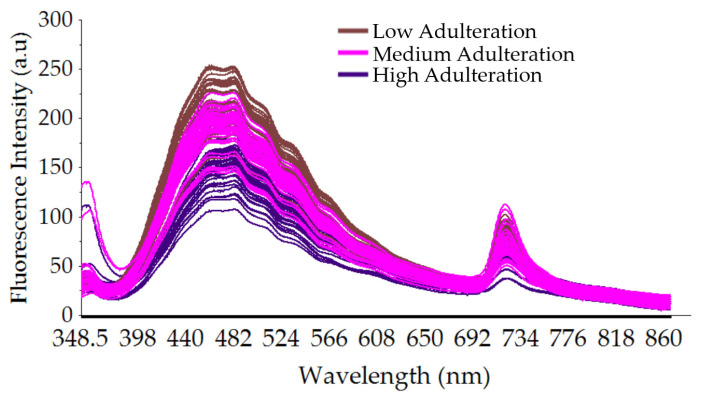
The smoothed fluorescence spectral data of adulterated SBH with three adulteration levels (low, medium, and high adulteration) at a full spectrum of 348.5–866.5 nm.

**Figure 6 foods-13-03648-f006:**
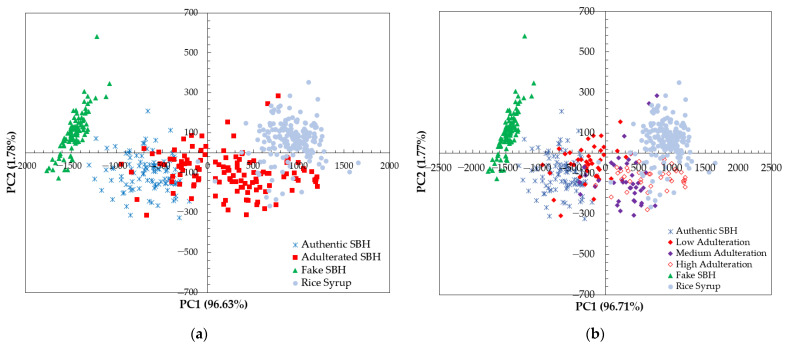
The result of PCA score plot calculation using a full spectrum of 348.5–866.5 nm (**a**) based on raw spectral data; (**b**) based on the smoothed spectral data.

**Figure 7 foods-13-03648-f007:**
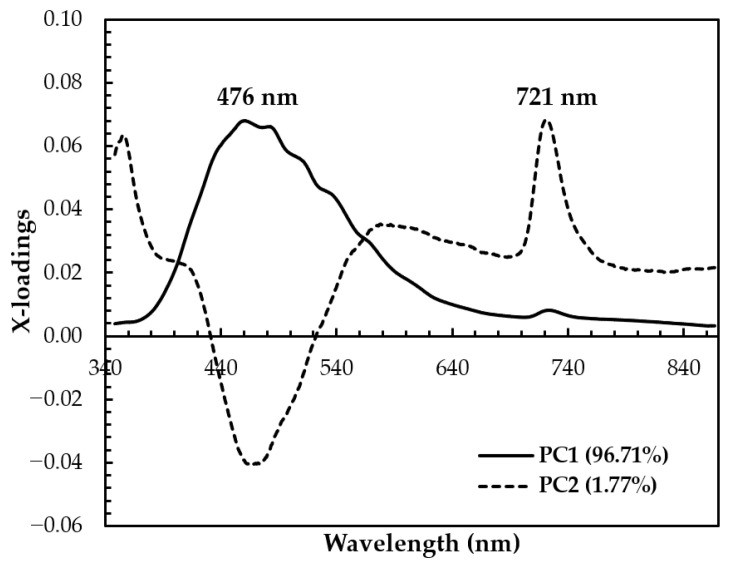
The calculated wavelength versus x-loadings for the first two PCs using a full spectrum of 348.5–866.5 nm.

**Figure 8 foods-13-03648-f008:**
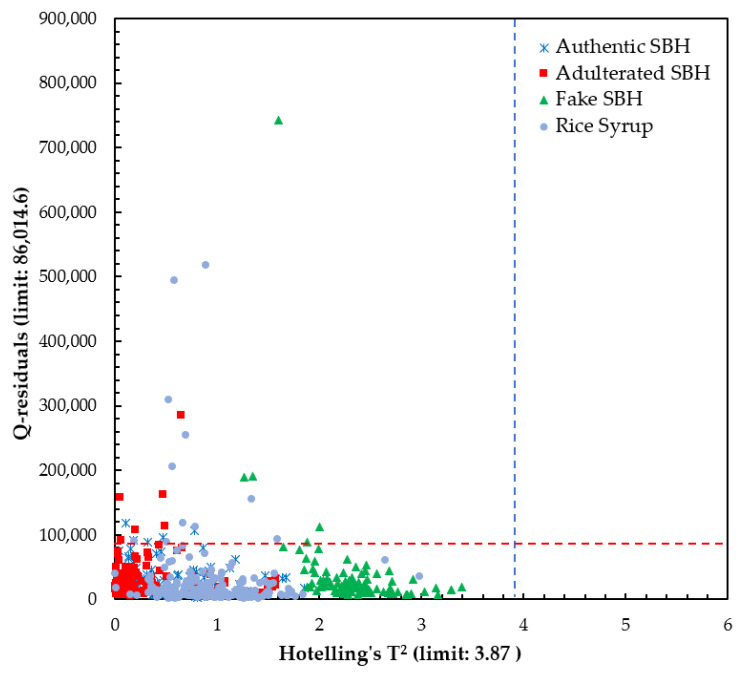
The calculated Hotelling’s T^2^ versus Q-residual using a full spectrum of 348.5–866.5 nm.

**Figure 9 foods-13-03648-f009:**
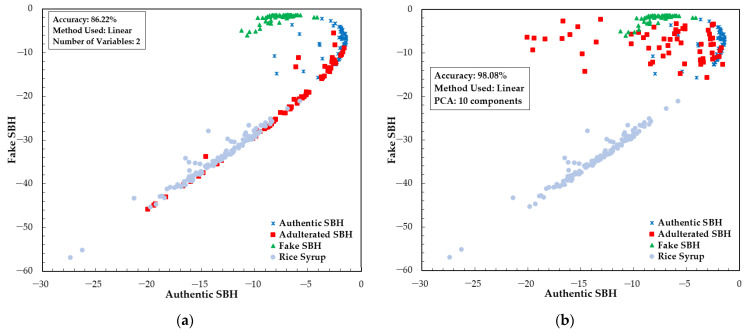
The result of classification model development with fewer selected variables: (**a**) the LDA method; (**b**) the PCA-LDA method.

**Figure 10 foods-13-03648-f010:**
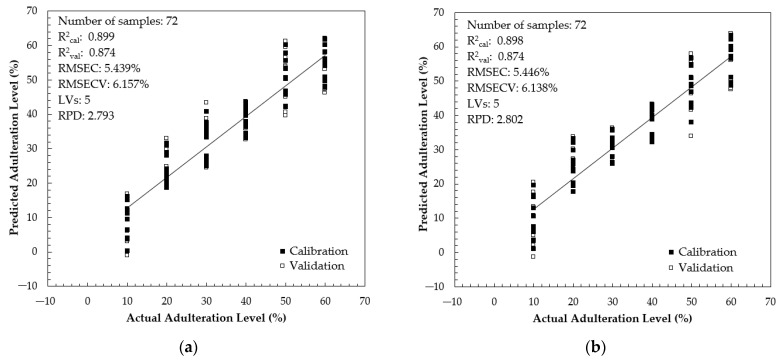
The scatter plot between actual and predicted adulteration levels in calibration and validation: (**a**) full-spectrum PLS model; (**b**) combined-interval PLS model.

**Figure 11 foods-13-03648-f011:**
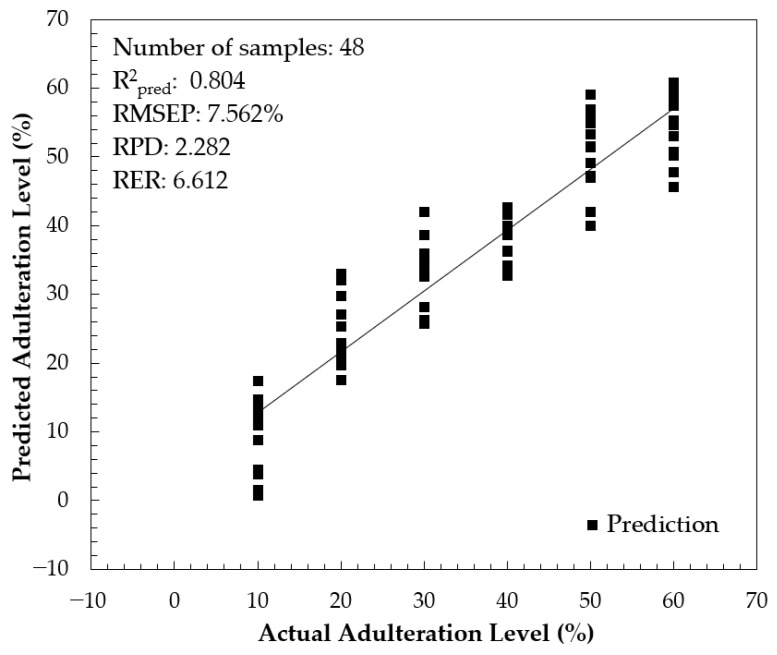
The scatter plot between actual and predicted adulteration levels.

**Table 1 foods-13-03648-t001:** SIMCA model development for authentic SBH, adulterated SBH, fake SBH, and rice syrup class using calibration and validation sample set.

SIMCA Model	Calibration and Validation Samples	Principal Components (PCs)	The Cumulative Percentage Variance (CPV) (%)	
			Calibration	Validation
Authentic SBH	60	3	98.3490	98.1350
Adulterated SBH	72	4	99.3491	99.2453
Fake SBH	60	4	99.0731	98.8279
Rice Syrup (RS)	120	4	98.8600	98.7172

**Table 2 foods-13-03648-t002:** Prediction results using four different classification models.

Model		Samples	Actual				Accuracy
			Authentic SBH	Adulterated SBH	Fake SBH	Rice Syrup	
SIMCA	Predicted	Authentic SBH	19	4	0	0	78.1%
		Adulterated SBH	20	39	0	19	
		Fake SBH	0	0	39	0	
		Rice Syrup	0	0	0	56	
PLS-DA	Predicted	Authentic SBH	30	0	0	0	86.5%
		Adulterated SBH	10	35	2	0	
		Fake SBH	0	13	38	3	
		Rice Syrup	0	0	0	77	
LDA	Predicted	Authentic SBH	36	11	0	1	85.6%
		Adulterated SBH	2	25	0	2	
		Fake SBH	2	0	40	0	
		Rice Syrup	0	12	0	77	
PCA-LDA	Predicted	Authentic SBH	40	0	0	0	99.5%
		Adulterated SBH	0	48	1	0	
		Fake SBH	0	0	39	0	
		Rice Syrup	0	0	0	80	

**Table 3 foods-13-03648-t003:** The result of PLS model development using full-spectrum and selected intervals with the NIPALS algorithm and the full-cross-validation method.

Intervals	Region	R^2^	RMSEC (%)	RMSECV (%)	RPD	RER
Full spectrum	348.5–866.5 nm	0.899	5.439	6.157	2.793	8.121
1	348.5–398.0 nm	0.823	7.189	7.821	2.199	6.393
2	398.5–448.0 nm	0.844	6.750	7.247	2.373	6.899
3	448.5–498.0 nm	0.873	6.082	6.567	2.619	7.614
4	498.5–548.0 nm	0.873	6.077	6.654	2.585	7.514
5	548.5–598.0 nm	0.871	6.138	6.821	2.521	7.330
6	598.5–648.0 nm	0.824	7.172	7.446	2.310	6.715
7	648.5–698.0 nm	0.795	7.727	8.128	2.116	6.152
8	698.5–748.0 nm	0.824	7.167	8.874	1.938	5.634
9	748.5–798.0 nm	0.764	8.305	12.642	1.360	3.955
10	798.5–866.5 nm	0.684	9.607	11.134	1.545	4.491

## Data Availability

The original contributions presented in the study are included in the article, further inquiries can be directed to the corresponding author.

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
