# Peer review of "Rapid Authentication of Intact Stingless Bee Honey (SBH) by Portable LED-Based Fluorescence Spectroscopy and Chemometrics"

_foods, 2024, doi:10.3390/foods13223648_

Round 1
Reviewer 1 Report
Comments and Suggestions for Authors
This manuscript reports a method for the quantitative and qualitative analysis of syrup adulteration in honey. The manuscript still has the following issues that need further clarification.
1. Necessary literature research needs to be expanded in the preface of the manuscript. The research on quantitative adulteration of honey has been extensively reported and the author needs to explore it in more depth.
2. As shown in Figure 1 and Figure 2, there are already significant differences in colour between different samples. Can we directly determine the approximate content of such adulterants in practical applications based on colour, without the need for such complex research steps as in the manuscript?
3. Suggestions for some figures in the manuscript. Annotation and explanation of characteristic peaks in Figure 3. Figure 6 can further investigate whether PCA can show better performance in 3D mode.
4. The optimisation methods for modelling feature variables can be further explored using other methods. Traditional band extraction seems to have been extensively studied and confirmed to be an inefficient task.
5. Furthermore, the models used for quantitative or qualitative analysis in current manuscripts seem to be somewhat outdated from a modelling point of view. Some new and efficient models are encouraged to be tried in current research data analysis tasks.
Reviewer 2 Report
Comments and Suggestions for Authors
The manuscript entitled "Rapid authentication of intact stingless bee honey (SBH) by portable LED-based fluorescence spectroscopy and chemometrics" deals with always up to date topic of food adulterations. Since cheap adulterations can have various negative effects on human health I find this manuscript very interesting. Also, rapid authentication methods are well-welcomed and necessary in order to improve quality and overall work in industry. All relevant references were used with a few instances noted in the comments where references should be added. The manuscript is thoroughly prepared regarding the references covering the results part and comparison with previously published results. All derived conclusions arise from the well-discussed presented results. The manuscript has scientific importance as well the importance for the food industry. I suggest that paper should be accepted after following minor corrections:
1. Abstract: line 23: Remove for samples.
2. Abstract: line 23: Change were to was.
3. Abstract: line 30: ... was 99.5% AND could...
4. Key words: Perhaps the number of keywords can be reduced.
5. Introduction, line 45: Word tremendous is not appropriate for scientific literature please rephrase.
6. Introduction, line 46: Excellent biological properties, please list them and reference them.
7. Introduction, line 53: Some chronic diseases, please list them and reference them.
8. Introduction, lines 58 and 62: Perhapse C4, CAM and C3 plants should be explained in brackets to make it easier for readers.
9. Introduction, lines 60: More than 7%, please add a reference.
10. Introduction, lines 61: SCIRA also has some limitations, please list them and reference them.
11. Results and discussion, line 174: The four samples THAT have...
12. Results and discussion, Figure 9, Describe picture in the terms of samples clustering, this part should be explained in more detailed way.
13. Results and discussion, line 359 and Table 3: Please even out the number of decimal places
14. Conclusion: The conclusion must include the most important results.
Reviewer 3 Report
Comments and Suggestions for Authors
Dear author the article is well written and interesting.
you can find the revision in the attached file.
